# Sex Differences in Cochlear Transcriptomes in Horseshoe Bats

**DOI:** 10.3390/ani14081177

**Published:** 2024-04-14

**Authors:** Jianyu Wu, Can Duan, Linjing Lan, Wenli Chen, Xiuguang Mao

**Affiliations:** School of Ecological and Environmental Sciences, East China Normal University, Shanghai 200062, China; 51253903017@stu.ecnu.edu.cn (J.W.); 51263903023@stu.ecnu.edu.cn (C.D.); ljlanbioo@gmail.com (L.L.); 51193904033@stu.ecnu.edu.cn (W.C.)

**Keywords:** gene expression, alternative splicing, RNA-seq, phenotypic variation, horseshoe bats

## Abstract

**Simple Summary:**

Phenotypic difference between males and females (sexual dimorphism) is widespread in animals. These sexual dimorphisms, in particular vocalizations and acoustic signals, have been shown to play important roles in mating choice and sexual selection. However, little is known about the molecular mechanisms underlying these phenotypic variations. In this study, we used the four taxa of the horseshoe bats to explore the patterns of gene regulations responsible for sex differences of echolocation pulse frequency. By analyzing the transcriptomes of both males and females in each taxon, we identified the genes with either a differential expression or alternative splicing and some of these genes are found to be related to hearing in human or mice. Our results support that differences in the degree of phenotypic sexual dimorphism can be encoded by the magnitude of sex-biased gene expression or splicing. In addition, our results indicate that the sex differences of the echolocation pulse may contain multiple parameters apart from the frequency component. Overall, this study provides growing evidences for supporting the important roles of both gene expression changes and alternative splicing in phenotypic variations.

**Abstract:**

Sexual dimorphism of calls is common in animals, whereas studies on the molecular basis underlying this phenotypic variation are still scarce. In this study, we used comparative transcriptomics of cochlea to investigate the sex-related difference in gene expression and alternative splicing in four *Rhinolophus* taxa. Based on 31 cochlear transcriptomes, we performed differential gene expression (DGE) and alternative splicing (AS) analyses between the sexes in each taxon. Consistent with the degree of difference in the echolocation pulse frequency between the sexes across the four taxa, we identified the largest number of differentially expressed genes (DEGs) and alternatively spliced genes (ASGs) in *R. sinicus*. However, we also detected multiple DEGs and ASGs in taxa without sexual differences in echolocation pulse frequency, suggesting that these genes might be related to other parameters of echolocation pulse rather than the frequency component. Some DEGs and ASGs are related to hearing loss or deafness genes in human or mice and they can be considered to be candidates associated with the sexual differences of echolocation pulse in bats. We also detected more than the expected overlap of DEGs and ASGs in two taxa. Overall, our current study supports the important roles of both DGE and AS in generating or maintaining sexual differences in animals.

## 1. Introduction

Sexual dimorphism, defined as phenotypic difference between males and females, is widespread in sexual organisms, such as the size and plumage of red-winged blackbird (*Agelaius phoeniceus*) and the horns of bighorn sheep (*Ovis canadensis*). The sexual dimorphism of vocalizations and acoustic signals is common in animals, such as in fishes [1], frogs [2], birds [3,4,5], and mammals [6,7]. This sexual differences have been shown to play important roles in mating choice and sexual selection [7,8,9].

In bats, echolocation is primarily used to negotiate the environment and to detect prey [10]. During echolocation, bats emit high echolocation pulse frequency which must be perceived as a pulse-echo combination [11] by the same individual to effectively extract information of the target [12]. Sex differences in echolocation pulse frequency have been found in many bat species [13,14]. Studies of echolocation pulses usually focused on the cochlea which is essential for hearing and echolocation [15,16,17,18]. Horseshoe bats (*Rhinolophus*) comprise over 100 recognized species [19] and emit a constant frequency (CF) in echolocation pulses which can be accurately assessed by researchers [14]. In addition, echolocation pulses emitted by horseshoe bats match the ‘acoustic fovea’ of the hearing system, meaning that, in these bats, hearing uses the same frequency as echolocation pulses [13,20]. Multiple horseshoe bats (*Rhinolophus*) have been reported to show sexual dimorphism in echolocation pulse frequency (the predominant constant frequency), including *R. hipposideros* [21], *R. blasii* [14], *R. pumilus* [22], *R. monoceros* [23], *R. pusillus* [24], *R. ferrumequinum* [25], *R. rouxi* [26], and *R. sinicus* [27,28]. It was notable that no sexual differences in echolocation pulse frequency were found in several horseshoe bats, such as *R. euryale* and *R. mehelyi* [14,29], as well as *R. affinis* [30].

Sexually dimorphic traits can be mediated by differential gene expression (DGE) between the sexes [31,32], such as male sexual dimorphism in turkeys [33], ornamental coloration in guppies [34], and nuptial spines in toads [35]. In addition, alternative splicing (AS), as an alternative form of gene regulation, has also been proved to be important in generating sexually dimorphic traits [36,37,38]. Several recent studies have been performed to assess the relative roles of DGE and AS in sexual differences and their results suggested that DGE and AS might function independently to mediate sexual differences [39,40]. However, few studies have been performed to explore the molecular mechanisms underlying the sexual differences of vocalizations and acoustic signals (but see [41]) and none have been conducted for the differences in echolocation pulse frequencies between males and females in bats.

In this study, we aim to identify and characterize candidate genes with expression changes or alternative splicing associated with sexual differences in echolocation pulse frequencies in bats. For this aim, we chose four *Rhinolophus* taxa as the study system, including *R. sinicus* and *R. pusillus* showing the sexual dimorphism in echolocation pulse frequency [24,27,28], as well as two subspecies of *R. affinis* (*R. affinis himalayanus* and *R. affinis hainanus*) that exhibit no sexual differences in echolocation pulse frequency [30]. The latter two taxa can be used as the controls compared to the former two taxa. Based on cochlear transcriptome sequencing (RNA-seq) data, we conducted DGE and AS analyses to identify the differentially expressed genes (DEGs) and alternatively spliced genes (ASGs) between the males and females in each of four taxa. If there was a correlation between the sex-biased gene expression and the difference of the degree of sexual dimorphism, as documented previously in the wild turkey (*Meleagris gallopavo*) [33], we predict that there would be more DEGs or ASGs identified between the sexes in *R. sinicus* and *R. pusillus* than in *R. affinis himalayanus* and *R. affinis hainanus*.

## 2. Materials and Methods

### 2.1. Study System and Tissue Collection

In this study, we studied the four taxa of horseshoe bats (*R. sinicus*, *R. pusillus*, *R. affinis himalayanus*, and *R. affinis hainanus*) and for each taxon, we captured adult bats of each sex from a single population (Appendix A and Figure 1a). For each bat, we used Avisoft UltraSoundGate 116Hnb kit (Avisoft, Berlin, Germany) to record its echolocation pulses which were analyzed using BatSound Pro version 3 (Fast Fourier Transformation size 1024, Hanning window). The predominant constant frequency of the second harmonic was extracted as the echolocation pulse frequency of each bat. All bats were euthanized by cervical dislocation. Cochleae of each bat were collected (Figure 1a) and frozen immediately in liquid nitrogen. Tissue were stored at a −80 °C freezer before RNA extraction.

### 2.2. RNA Extraction, Library Construction, Sequencing, and Mapping

For each sample, the total RNA was extracted using TRIzol (Life Technologies Corp., Carlsbad, CA, USA) and libraries were constructed using TruSeq mRNA Standard library preparation kit (Illunima, San Diego, CA, USA). All libraries were qualified by Agilent 2100 Bioanalyzer and sequenced using Illumina HiSeq X Ten (paired-end 150 bp). Raw reads from each sample were trimmed using TRIMMOMATIC version 0.36 [42] with a sliding window of 4 bp, a minimum average PHRED quality score of 20, and minimum reads length of 50 bp. Detailed information about sequencing reads for each sample has been provided in supporting information (Appendix A).

For each taxon, the clean reads of each sample were aligned with the respective high-quality reference genome, except for *R. pusillus* ([43] and Appendix A) using HISAT2 v2.2.0 [44] with default parameters. For *R. pusillus* samples, we used *R. rex*, which is closely related to *R. pusillus*, as the reference (Appendix A). SAMtools v1.11 [45] was applied to generate sorted BAM files, and the mRNA alignments in sorted BAM files were used in following analyses.

### 2.3. Differential Expression Analysis

FeatureCounts v2.0.1 [46] was used to quantify the mapped reads in the alignment and read count matrix across samples was normalized using DESeq method in DESeq2 v1.30.1. [47]. The possible batch effect was adjusted using the sva function implemented in R package SVA [48]. Then, we used R package (R-Core-Team 2015) to perform a principal component analysis (PCA) to explore the similarity of expression patterns across all the samples of each taxon.

Prior to DGE analysis, the low expressed genes with an average count per million (CPM) <1 across samples in each taxon were filtered out. Then, we used DESeq2 [47] to identify differentially expressed genes (DEGs) between the sexes of each taxon with the *p* value < 0.05 after Benjamini and Hochberg adjustment for multiple tests [49] (padj < 0.05). To reduce the false positives, we also filtered out those genes with |log_2_(fold change (FC))| > 0.5.

### 2.4. Alternative Splicing Analysis

We applied DEXSeq v 1.42.0 [50] to determine differential exon usage (DEU) between the sexes of each taxon and genes with significance were identified as alternatively spliced genes (ASGs). We first used the Python script ‘dexseq_prepare_annotation2.py’ in DEXSeq package to flatten the genome annotation file and then the Python script ‘dexseq_count.py’ was used to quantify exon-specific read counts, resulting in the count table for each sample. Following the DGE analysis using DESeq2 above, we determined significant differences in exon usage between the sexes with |log_2_FC| > 0.5 and padj < 0.05.

### 2.5. Comparison of Differential Expression and Alternative Splicing

To test whether differential gene expression and alternative splicing act independently in gene regulation, we assessed the extent of overlap between the DEGs and ASGs identified in each taxon. Following previous studies [38,40], we first calculated the expected number of genes that are both DEGs and ASGs as (total no. DEGs × total no. ASG)/total no. expressed genes. Then, we used the representation factor (RF, the ratio of the observed number of overlapped genes and the expected number) to assess the extent of overlap (RF > 1: more overlap than expected; RF < 1: less overlap than expected). Significance of RF was determined using a hypergeometric test in R version 4.0.5 with the cut-off *p*-value of 0.05.

### 2.6. Functional Enrichment Analysis

To characterize the functional role of DEGs and ASGs, we performed Gene Ontology (GO) term enrichment analysis using ShinyGo v0.77 [51] with an FDR cutoff value of 0.05. The top 20 significantly enriched GO Biological Processes were shown in plots.

## 3. Results

### 3.1. Characterization of Echolocation Pulse Frequency Variation across the Four Rhinolophus Taxa

Consistent with the previous studies in *R. sinicus* [27,28] and *R. pusillus* [24], we found significant differences in the echolocation pulse frequency between males and females in these two species (Appendix A and Figure 1b). In addition, we also confirmed the previous results in two subspecies of *R. affinis* (*R. affinis himalayanus* and *R. affinis hainanus*) that no sexual differences in the echolocation pulse frequency was observed [30] (Appendix A and Figure 1b).

### 3.2. Cochlear RNA-Seq Data Collection and Mapping

We generated cochlear transcriptomic data from 31 samples of four *Rhinolophus* taxa (Appendix A) with an average of 46 million clean paired reads per sample (Appendix A). The clean reads of each sample in each taxon were mapped to the respective high-quality reference genome with an overall alignment rate of 92.78% (Appendix A). The mRNA alignments in sorted BAM files were used in both differential expression (DE) and alternative splicing (AS) analyses (Figure 1a).

### 3.3. Characterization of Sex-Biased DEGs in Cochlea of Four Rhinolophus Taxa

We found a batch effect in the RNA-seq data of each taxon revealed by principal component analysis (PCA) (Appendix A). After adjusting batch effect using R package SVA, samples of each sex in each taxon separated from each other clearly in the PCA plots (Appendix A). Based on SVA-adjusted expression matrix, we performed DGE analysis between males and females in each taxon. In general, we found a higher number of DEGs in *R. sinicus* and *R. affinis himalayanus* (405 and 301 in the former and latter, respectively) than in *R. pusillus* and *R. affinis hainanus* (46 and 43 in the former and latter, respectively) (Figure 2a, Appendix A, and Appendix A). In addition, the number of female-biased genes was more than that of male-biased genes in all of the four taxa except for *R. affinis himalayanus* (Figure 2a).

Specifically, in *R. sinicus,* we identified the largest number of DEGs with 136 male-biased and 269 female-biased genes (Appendix A). Functional enrichment analysis revealed that male-biased genes are enriched in GO terms related to ion transmembrane transport, muscle system process, central nervous system development, and cellular component morphogenesis, whereas female-biased genes are enriched in GO terms associated with cytoplasmic translation, mitotic cell cycle process, and mitochondrial ATP synthesis coupled electron transport (Appendix A and Appendix A). In *R. pusillus,* we identified 19 male-biased and 27 female-biased genes (Appendix A). These male-biased genes are enriched in GO terms related to carboxylic acid metabolic process, whereas female-biased genes are enriched in GO terms associated with ion transport, transmembrane transport, enteric nervous system development, and defense response to fungus and bacterium (Appendix A and Appendix A).

Unexpectedly, in *R. affinis himalayanus* with no sexual differences of echolocation pulse frequencies, we also identified a large number of DEGs with 257 male-biased and 44 female-biased genes (Appendix A). Only male-biased genes are functionally enriched in GO terms that are related to immune function and cytoskeleton organization (Appendix A and Appendix A). In contrast, in another subspecies of *R. affinis* (*R. affinis hainanus*), we identified a small number of DEGs with 10 male-biased and 38 female-biased genes (Appendix A) and again only male-biased genes are enriched in GO terms that are related to elastic fiber assembly and the regulation of immune response (Appendix A and Appendix A).

To further identify genes whose expression changes are associated with sexual differences in the echolocation pulse frequency, we used a candidate gene approach by comparing the sex-biased DEGs identified in each taxon to the list of hearing loss or deafness genes obtained from the database of HMDC (The Human–Mouse: Disease Connection, accessed on 18 January in 2024). Although these candidate hearing loss or deafness genes are not enriched in these sex-biased DEGs in each taxon (all *p* > 0.05 in a hypergeometric test, Appendix A), we still found multiple such hearing loss or deafness genes in the list of DEGs in each taxon, ranging from four in *R. pusillus* to 33 in *R. sinicus* (Figure 2e–h and Appendix A). Specifically, in *R. sinicus*, *POU1F1* shows the largest fold change except for two Y-linked genes (Appendix A) and its protein is the first pituitary-specific transcription factor identified in the human and mouse with a restricted expression in the anterior pituitary lobe [52]. The male-biased expression of *POU1F1* observed in this study suggests its possible role in the development of the sexual difference of inner ear. Another notable one is *OTOS* which is highly expressed in the fibrocytes of the inner ear and the downregulation of this gene can cause irreversible deafness with the severe degeneration of hair cells [53,54]. This gene has also been identified as a candidate echolocation gene associated with the convergence of echolocation between bats and wales [55]. Previous RNA-seq data from cochlear tissue have also shown that this gene (*OTOS*) and *CEACAM16* were found to be significantly upregulated in echolocating bats relative to non-echolocating bats [56], as well as upregulated in constant frequency (CF) echolocating bats relative to non-CF bats [15].

To investigate whether sex-biased gene expression is conserved across species, we compared the list of DEGs identified in each taxon with each other and found several overlapped DEGs between taxa (Figure 2b–d and Appendix A). To further identify those genes which are male-biased in one taxon but female-biased in another taxon or the reverse, we compared the combined male-biased genes in all four taxa to those of female-biased genes and found 22 such genes including *LRP2* (Appendix A and Appendix A).

### 3.4. Characterization of Sex-Biased ASGs in Cochlea of Four Rhinolophus Taxa

Similarly to the number of DEGs identified in each taxon above, we also found the largest number of ASGs in *R. sinicus* and the lowest in *R. affinis hainanus* (Figure 3a, Appendix A and Appendix A). Functional enrichment analysis revealed that significant GO terms were only found on the ASGs in *R. sinicus* and they are related to cellular homeostasis, RNA processing, and RNA splicing (Appendix A and Appendix A). It was notable that only two genes were identified as ASG between males and females in *R. affinis hainanus* (Appendix A). One of them was not functionally annotated. Another one (*PHB2*) is an inner mitochondrial membrane mitophagy receptor [57] and has been proven to be involved in age-related hearing loss in mice [58].

Similarly to the case in the DEGs above, candidate hearing loss or deafness genes are also not enriched in these ASGs between the sexes in each taxon (all *p* > 0.05 in a hypergeometric test, Appendix A). However, we still found 19, 11, and 1 hearing loss or deafness genes in the ASGs of *R. sinicus*, *R. affinis himalayanus,* and *R. pusillus*, respectively (Appendix A). Some of these hearing loss or deafness genes have also been identified as candidate echolocation genes, such as *CDH23* [59] and *LRP2* [60].

Unlike the DEGs above, we found only one ASG overlapped between species (*LGALS8* between *R. sinicus* and *R. affinis himalayanus*, Figure 3b). This gene in humans has been shown to encode seven different isoforms due to alternative splicing and express widely in tumor tissues [61]. However, little is known about the function of this gene in cochlea.

### 3.5. Comparisons of Differential Expression and Alternative Splicing

To investigate whether differential expression and alternative splicing function independently in gene regulation, we assessed the extent of overlap between them. No overlapping genes between DEGs and ASGs were found in *R. affinis hainanus* and *R. pusillus* due to a small number of both DEGs and ASGs in these two taxa. In the other two taxa (*R. affinis himalayanus* and *R. sinicus*), we found greater overlap than expected between DEGs and ASGs, although significance was only detected in the former (*R. affinis himalayanus*: RF = 4.10, *p* = 0.0002; *R. sinicus*: RF = 1.352, *p* = 0.1041) (Figure 4). Among the overlapped DEGs and ASGs, we found two and one hearing loss/deafness genes in *R. affinis himalayanus* (*MMP9* and *MPO*) and *R. sinicus* (*TPM2*), respectively (Appendix A).

## 4. Discussion

In this study, we used four *Rhinolophus* taxa as the study system, including two taxa (*R. sinicus* and *R. pusillus*) with a sexual dimorphism of the echolocation pulse frequency and two other taxa (*R. affinis himalayanus* and *R. affinis hainanus*) without sexual differences in the echolocation pulse frequency to investigate the molecular basis underlying the sexual dimorphism of the echolocation pulse frequency. With the RNA-seq data of cochlear tissues from males and females, we conducted both differential gene expression (DGE) and alternative splicing (AS) analyses.

Consistent with the largest degree of sexual difference in the echolocation pulse frequency observed in *R. sinicus*, we detected the largest number of differentially expressed genes (DEGs) between the sexes in this species compared to the three other taxa. This seems to support that differences in the degree of phenotypic sexual dimorphism between closely related species can be encoded by the magnitude of sex-biased gene expression [33]. However, we also identified multiple sex-biased genes in two subspecies of *R. affinis* which have no difference in echolocation pulse frequency between the sexes, particularly in *R. affinis himalayanus*. A behavioral study of *R. euryale* and *R. mehelyi*, another two horseshoe bats with no sexual dimorphism in echolocation pulse frequency [14,29], has shown that these two species can recognize the sex of conspecifics from echolocation pulses [62]. This suggested that the constant frequency component of echolocation pulses might not be the sex-specific acoustic signal. Thus, the sex-biased genes identified in the two subspecies of *R. affinis* here may be more likely associated with other parameters of echolocation pulses rather than the constant frequency component. Consistent with this suggestion, the sexual dimorphism of echolocation pulses has been observed in other multiple pulse parameters apart from the predominant constant frequency in CF-FM bats [63]. Alternatively, these genes might contribute to the constant frequency component indirectly, possibly by co-expression in gene regulatory networks [64].

In this study, we found several overlapped sex-biased genes across the four taxa (Figure 2b), suggesting that sex-biased gene expression may be conserved in recently diverged taxa. Specifically, among female-biased genes across taxa, *SLC6A2* overlapped between *R. pusillus* and *R. sinicus*, which is one of the ADHD risk genes in humans and has also been identified to have sexually dimorphic effects with a greater effect on females than on males [65]. Another notable gene is *LRP2* overlapped between *R. pusillus* and *R. affinis hainanus*, which was identified as a deafness gene [66,67] as well as a candidate echolocation gene associated with the origin of laryngeal echolocation in bats [60]. A previous study also showed that *LRP2* can partially mediate a female-specific effects of *LCN2* on metabolic traits in mice [68]. Interestingly, this gene was identified to be male-biased in *R. sinicus*. In contrast to the findings herein, a study of *Heliconius* butterflies revealed that sex-biased gene expression occurred in a species-specific manner [69]. Thus, the evolution of gene expression between the sexes may depend on diverse factors, including not only phylogeny but also the pressure of either sexual selection or natural selection or both [70].

Consistent with the results of a recent study based on the RNA-seq data of liver and brain [40], we found two Y-linked genes (*KDM5D* and *DDX3Y*) specifically expressed in males’ cochlear tissue in *R. sinicus*. In addition, *DDX3Y* was also detected to be a male-specific gene in *R. affinis hainanus*. The consistency across species and tissues supports an important role of Y chromosome genes in the formation of sexual dimorphic traits [71,72].

Compared to a recent study on sex differences in AS in the liver and brain of *R. sinicus* [40], we identified a smaller number of alternatively spliced genes (ASGs) in the cochlear tissue (over 1000 in [40] and less than 250 in this study, see Figure 3a). This mainly resulted from the different methods used between studies (rMATs in [40]; DEXSeq in this study). Another recent study that used both rMATs and DEXSeq when performing AS analysis in the same tissue revealed that the former method detected much more ASGs than the latter [73] (1932 with rMATs and 1267 with DEXSeq). Nevertheless, our current study and [40] in diverse bat species and tissues support an important role of AS in encoding sexual differences [36,39,74,75].

Due to a small number of ASGs detected in *R. pusillus* and *R. affinis hainanus*, the overlapping analysis was only conducted in *R. sinicus* and *R. affinis himalayanus*, which revealed a greater overlap than expected between DEGs and DSGs in both species, although this was not significant in the latter (Figure 4). Our current results from cochlear tissue were consistent with the result from the brain but not from the liver in *R. sinicus* [40]. As suggested in [40], this inconsistency might be caused by tissue- and species-specific gene expression and splicing [38,39]. Alternatively, this might have resulted from the difference in function across tissues. Specifically, the function of brain and cochlea is closely related during the process of echolocation [76], which might explain why similar expression patterns were observed in these two tissues. Compared to patterns of sex-biased gene expression across the four *Rhinolophus* taxa, we observed less conservation of sex-biased splicing across species. This may be caused by the unique properties of each of these two gene regulation forms [77]. In addition, a higher rate of evolution in ASGs than DEGs may provide another scenario to explain the less conservation of AS than DGE across species [38].

## 5. Conclusions

By performing the differential expression and alternative splicing analyses on the transcriptomes of cochlea in four *Rhinolophus* taxa, we identified some DEGs or ASGs related to hearing loss or deafness in human or mice (e.g., *OTOS*, *CEACAM16*, *LRP2*, and *CDH23*), and these genes might be associated with sexual difference of echolocation pulses in bats. This study is the first to investigate the molecular basis of high-frequency hearing differences between the sexes using the transcriptomes of cochlea, and our results support the important roles of both DGE and AS in phenotypic variations.

## Figures and Tables

**Figure 1 animals-14-01177-f001:**
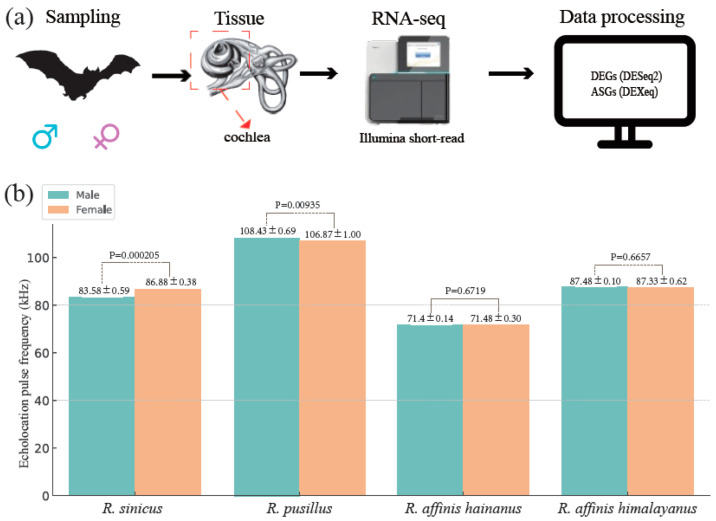
Experimental design (**a**) and variations of echolocation pulse frequency between males and females across the four taxa (**b**). Significance of the difference between males and females was determined using the T-test.

**Figure 2 animals-14-01177-f002:**
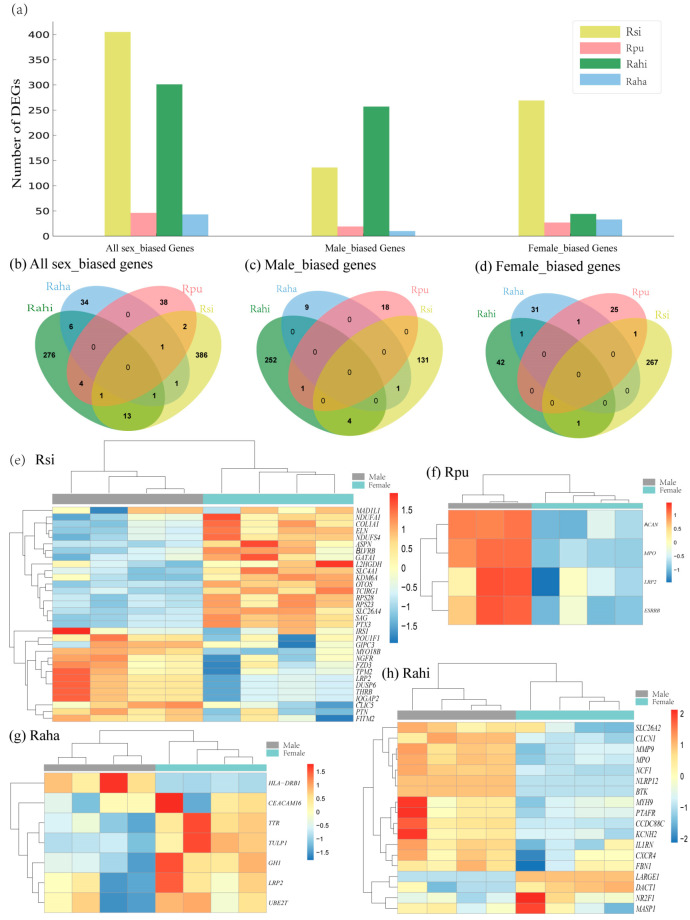
Differential expression analysis between males and females in each taxon. (**a**) Histograms showing the number of differentially expressed genes (DEGs) in each taxon, including all sex-biased genes, male-biased genes, and female-biased genes. (**b**–**d**) Venn diagrams showing the number of all sex-biased genes (**b**), male-biased genes (**c**) and female-biased genes (**d**) across the four taxa. (**e**–**h**) Hierarchical clustering and heatmaps showing expression patterns of hearing loss/deafness genes among DEGs in Rsi (**e**), Rpu (**f**), Rahi (**g**), and Raha (**h**), respectively. Rsi: *R. sinicus*, Rpu: *R. pusillus*, Rahi: *R. affinis himalayanus*, Raha: *R. affinis hainanus*.

**Figure 3 animals-14-01177-f003:**
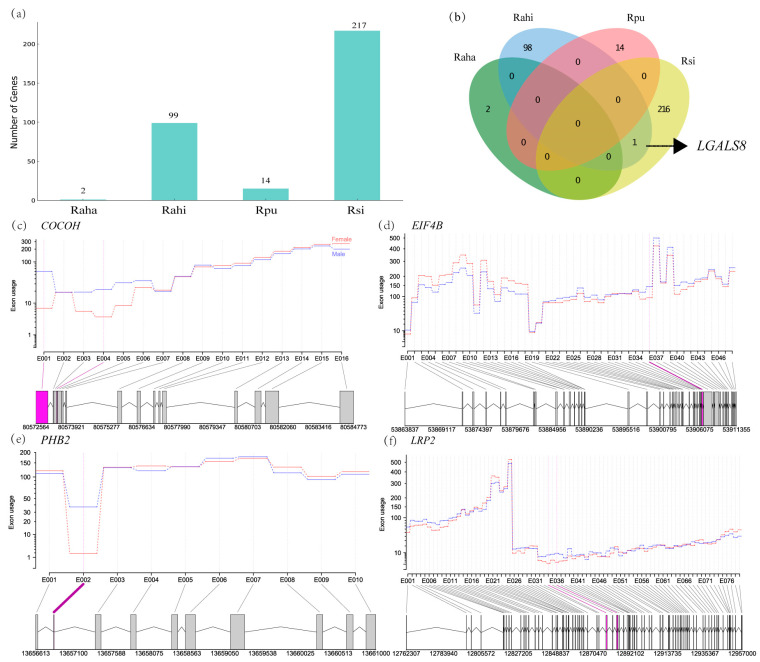
Alternative splicing analysis between males and females in each taxon. (**a**) Histograms showing the number of alternatively spliced genes (ASGs) between the sexes in each taxon. (**b**) Venn diagram showing the number of ASGs between the sexes across the four taxa. (**c**–**f**) Examples of ASGs identified between the sexes in each taxon using DEXSeq. Exons with significant differential usage between the sexes are shown in purple. Rsi: *R. sinicus*, Rpu: *R. pusillus*, Rahi: *R. affinis himalayanus*, Raha: *R. affinis hainanus*.

**Figure 4 animals-14-01177-f004:**
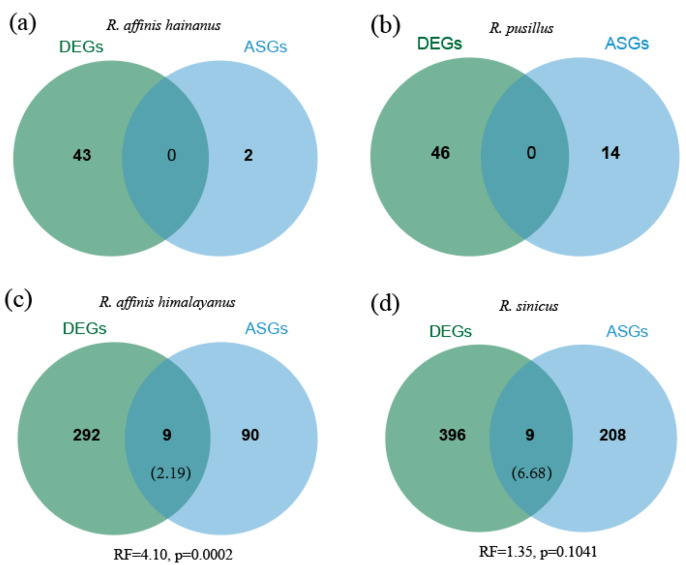
Overlap of DEGs and ASGs between sexes in *R. affinis hainanus* (**a**), *R. pusillus* (**b**), *R. affinis himalayanus* (**c**), and *R. sinicus* (**d**). The expected number of overlapped DEGs and ASGs is shown in brackets under the observed number.

## Data Availability

All sequencing reads generated in this study have been deposited to NCBI’s Sequence Read Archive database (SRA) under BioProject accession no. PRJNA1070387.

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
