# Peer review of "Sex Differences in Cochlear Transcriptomes in Horseshoe Bats"

_animals, 2024, doi:10.3390/ani14081177_

Round 1
Reviewer 1 Report
Comments and Suggestions for Authors
Overview
The current manuscript entitled “Sex Differences of Cochlear Transcriptomes in Horseshoe Bats” presents a comprehensive study aimed at unraveling the molecular mechanisms underlying the sexual dimorphism in call frequencies observed among horseshoe bats using comparative transcriptomics.
The current research identifies numerous DEGs and ASGs that correlate with the variations in call frequency between male and female bats. The current study also notes an unexpected overlap in DEGs and ASGs between taxa, indicating possible conservation or shared mechanisms across different species.
Overall, the objectives and rationale of the study are clearly stated, the statistical analysis and reporting is reasonable. However, there are a few issues with the experimental design and conclusion.
General comments and questions
Major
Please justify the relevance and inclusion of two R. affinis species in this study which do not exhibit sex difference in call frequency.
Some conclusions lack sufficient support and rigor. For example, the authors detected multiple DEGs and ASGs in taxa without sexual differences of call frequency, and concluded these genes are not relevant in the frequency component. This is not rigorous, these genes are still possible to contribute to the frequency component but in other indirect mechanisms, for example, co-expression regulation.
AS (alternative splicing) usually consists of DEU (differentially exon usage) and DIE (differential isoform expression analysis). The authors are encouraged to also perform DIE analysis in order to better characterize AS events and identify AS candidates.
Minor
The logic and information flow in the introduction section can be improved to ensure clarity and enhance the reader's understanding. A possible layout could be:
· Contextual opening: broad overview of sexual dimorphism, provide vivid examples of such phenomenon. By the end, transit to vocalizations and acoustic signals.
· Narrowing down: Elaborate on vocalizations and acoustic signals, for example, introduce the important role in mating choices and sexual selection, provide examples. Then transit to and elaborate on echolocation calls in bat.
· Technical Information: Briefly introduction sequencing technology advancement and omics approaches for investigating such biological questions.
· Identify gaps and mention current study’s relevance, stating the objectives and hypothesis.
Some content in the results section is actually discussion, for example, elaboration of specific genes in line 200 – 231. Considering moving them to the discussion section. As a consequence, the discussion section is not comprehensive.
Adding a volcano plot for the DEU analysis can help with the results presentation.
Specific comments and questions
Table 1: considering moving this table to supplementary files, also, try to simplify the table, by merging cells in species, sex, sampling, reference, etc. to avoid redundancy.
Figure2: try to match color annotation of each taxon in (a) and (b) to improve consistency and readability. Also, consider changing the y-axis label in (a) to “number of differentially expressed genes” to avoid confusion.
Line 62-66: this part is not easy to follow. consider simplify the logic.
Line 147: extra signs detected, need to be removed.
Line 167: “more number of” is not proper, should be “higher number of” for correct grammatical structure.
Comments on the Quality of English LanguageConsidering polishing the manuscript, focusing on simplifying and improve the logic of some long sentences.
Author Response
Please see the attachment。

Reviewer 2 Report
Comments and Suggestions for Authors
This study by Jianyu Wu uses RNA-Seq to search for and characterize gene expression and alternative splice differences in genes in the cochlea of two species of bats that have sex differences in call frequency (i.e. pitch) and two that are the same between males and females. They find sex differences in the cochlea of all four species, without a correlation of more differences in the species have production differences in their vocalizations. Some of the genes with sex differences have been identified as contributing to hearing disorders when disrupted. This study presents valuable and believable data, but more needs to be done to be more confident of some of the conclusions.
Line 120. For the genes filtered out with low expression, where these low in all samples or just at least one group or one sample? If at least one, then filtering out those genes could cause filtering out genes that are differential between the sexes
It would be good if the authors can confirm a few sex differences in gene expression and alternative splicing with RT-PCR analyses.
Figure 2. It would be good if the bar plots in panel (a) had the same color as the Venn diagram in panel (b).
Line 200. For the genes from the human-mouse disease database, the authors should do some statistical analyses to see if they are enriched in the genes with sex differences in bats. A hypergeometric test is one way of comparing statistical overlap between list.
Figure 3. Need to be clearer in the figure and figure legend that there are genes with alternative splice differences between the sexes. Some of the text makes it seem like the authors are saying simply that the cochlea has alternatively spliced differences, regardless of mentioning sex.
The text can use some improvement in English grammar. For example, in line 46, “few study” should be written as “few studies”.
Comments on the Quality of English LanguagePut comments above
Reviewer 3 Report
Comments and Suggestions for Authors
The topic of the work is very interesting: the potential differences between echolocation systems for calling frequencies in horseshoe bats. The introduction is very well presented, providing basic data on the concept of sexual dimorphism applied to molecular aspects related to differential gene expression. On the other hand, at the end of this introduction the objective of the research is clearly stated.
The methodology is clearly described and is appropriate to meet the objective of the work. It is important to use a figure that summarizes the experiment. It would only be important to incorporate the ethical aspects and corresponding authorizations into the description of animal handling.
The results are presented clearly and adequately graphed in figures.
In the Discussion I consider that it would be important to delve into some aspects, especially those related to dimorphism in gene expression of different organs. The authors mention that dimorphism has been found in the brain and not in the liver, which seems very easy to justify based on the functions of each organ. Furthermore, it also seems easy to clearly relate it to the sexual dimorphism in the gene expression of the cochlea. I consider that only by deepening these aspects of the discussion, the article is in a position to be published in Animals
Minor aspect:
Change by DGE the abbreviation of Differential gene expression
Round 2
Reviewer 1 Report
Comments and Suggestions for Authors
The manuscript has been sufficiently improved.
A minor point: in my reviewer version PDF, there are several random red horizontal lines throughout the figures, e.g., Figure 2 middle part, Figure 3 bottom, Figure 4 middle and bottom, could be formatting/editing issues. Please double check.
Author Response
A minor point: in my reviewer version PDF, there are several random red horizontal lines throughout the figures, e.g., Figure 2 middle part, Figure 3 bottom, Figure 4 middle and bottom, could be formatting/editing issues. Please double check.
>>> We have double checked all figures and we think that it might be formatting/editing issues.